# An Explanation and Defense of the Free-Thinking Argument

**Timothy A. Stratton** [1,*] **and J. P. Moreland** [2]

1    Trinity College of the Bible and Theological Seminary, Newburgh, IN 47630, USA
2    Talbot School of Theology, Biola University, La Mirada, CA 90639, USA
*    Correspondence: tim@freethinkinc.org

**Abstract:** This paper is a defense of the big ideas behind the free-thinking argument. This argument aims to demonstrate that determinism is incompatible with epistemic responsibility in a desert sense (being praised or blamed for any thought, idea, judgment, or belief). This lack of epistemic responsibility is problematic for the naturalist. It seems to be an even worse problem, however, for the exhaustive divine determinist because not only would humanity not stand in a position to be blamed for any of our thoughts and beliefs, but it also surfaces a "problem of epistemic evil", which can be raised against the knowledge of God, the rationality of humans, and the trustworthiness of Scripture.

**Keywords:** epistemic responsibility; libertarian freedom; determinism; deliberation

## 1. Introduction

Intrinsic value/disvalue and objective, normative duties, oughts/ought-nots are central to at least three areas of life: morality, rationality, and aesthetics. If one violates a moral ought, one is morally guilty. If one violates a rational ought, one is irrational. If one violates an aesthetic ought, one produces something ugly, or at least not as beautiful as if that ought had not been violated. In this paper, our focus is on rationality. We begin with what is called the free-thinking argument. After presenting and defending it, we draw out some important implications that follow from it.[1]

The free-thinking argument (hereafter, FTA) is part of a family of arguments that have been developed by thinkers such as C. S. Lewis, J. P. Moreland ([1987] 2000), Alvin Plantinga,[2] and Jim Slagle. It has evolved over the past decade as it has adapted to criticism. The big ideas behind the argument, however, have not changed.

The FTA seeks to discredit the thesis of determinism. Determinism is the idea that antecedent conditions are sufficient to necessitate all events or effects (often referred to as "causal determinism"). Exhaustive divine determinism (hereafter, EDD) is the idea that God necessitates all events—especially all things about humanity, which would include all desires, thoughts, intuitions, beliefs, actions, behaviors, evaluations, and judgments. It is important to note that antecedent conditions are either sufficient or insufficient to necessitate all effects.[3] With determinism in mind, the FTA is typically aimed at naturalism and the complete determinism of humanity which seems to be entailed by this view.[4] Naturalism, as Alvin Plantinga describes, is the view that neither God nor anything like God exists.[5] On other occasions, however, the argument is aimed at Calvinists who affirm EDD.[6]

## 2. The Free-Thinking Quiz

To feel the force of the argument, let us begin by answering a few questions. The first one is,

*Are we infallible?*

Everyone seems to know that he or she does not know everything. In fact, it seems that every mature and rational person is willing to admit that—at least on occasion—we affirm

false beliefs. That is to say, we know that each of us holds, affirms, and often advances beliefs that we really think are true and rationally justified, but are actually false.

This leads to the next question: At least sometimes, at a certain time and place, when we are affirming a false belief,

> *is it possible for us (in those same circumstances) to take our faulty thinking captive and think correctly?*

When we are affirming false beliefs, can we—that is, do we possess the power and opportunity to—act or think differently and be a bit more careful to reach true conclusions about reality in those specific circumstances?[7] The power to think and act—and be more careful than how something external to us determines us to be—seems to be a power worth wanting[8] and having, but how one answers these questions has significant ramifications on one's worldview. We contend that if we possess the power to—at least occasionally— evaluate our current thoughts and beliefs, rationally judge them as good, bad, right, or wrong, and then take our faulty thoughts captive and replace them with proper thoughts without change in antecedent conditions, then we possess libertarian freedom. This kind of freedom—that of deliberation—refers to a person's choice, action, evaluation, thought, or judgment that is not ultimately determined by something or someone else.[9] This implies that at least occasionally, we act as rational agents and freely form our judgments for which we are rationally responsible. This implication also entails that it is we ourselves, and not our overall antecedent mental states or character, that brings about our judgments.

Richard Taylor (1992) puts it this way: "In the case of an action that is free, it must not only be such that it is caused by the agent who performs it, but also such that no antecedent conditions were sufficient for his performing just that action". We can understand libertarian freedom, then, as the ability to choose such that antecedent conditions are insufficient to causally determine or necessitate one's choice. We have applied this definition to thinking: *Libertarian freedom includes the ability to think such that antecedent conditions are insufficient to causally determine or necessitate one's thoughts and ensuing beliefs.*[10]

This application of libertarian freedom to thinking holds whether or not one possesses alternative possibilities from which to choose (one simply needs to be the first mover of these choices and thoughts).[11] We refer to this idea as *weak libertarian freedom*. However, if one does face opportunities to choose among alternative possibilities in the real world (*strong libertarian freedom*) and has the power to choose among them with no changes in the mental conditions at the time of choice, then it follows that one is not determined by something or someone else.[12] We can understand strong libertarian freedom in this manner:

> The opportunity to exercise an ability to choose between at least two options, each of which is compatible with one's nature in a specific circumstance where the antecedent conditions are insufficient to causally determine or necessitate the agent's choice.

At the moment of choice, one can do A or B without any antecedent change within the agent's mental life. Each choice is up to the agent himself, and not the result of a change, e.g., in the agent's belief-desire set.

That is a fancy way of saying *the ability to do otherwise*. The ability to do otherwise is not necessary to have libertarian freedom. However, to assume a Christian biblical perspective, if one possesses this opportunity and ability to choose otherwise (as implied in 1 Cor 10:13), then this is sufficient to demonstrate libertarian freedom.[13] Ultimately, if one cannot demonstrate strong libertarian freedom, but can demonstrate weak libertarian freedom, one has demonstrated libertarian freedom and dispatched the idea of determinism.[14]

So, to reframe the last question:

> When engaged in deliberation and belief formation, do we sometimes possess the opportunity and ability to act, think, and be more careful when we are thinking things through, or does something or someone else always determine exactly

what we think of and about and more importantly, determine exactly how we think of and about it?

Are we ever epistemically responsible (in a desert sense) for any of our thoughts and beliefs, or is something or someone else ultimately responsible for all of our thoughts and beliefs?[15] In a process of deliberation, are we a genuinely responsible active agent who is steering the ship or are we passive patients, spectators, passengers simply along for the ride waiting to see what the antecedent causal chain that begins before the "deliberative" process and runs through our mental states will end up causing?

Much has been written on this topic, but to make it simple we would like to show that a kind of rationality worth wanting requires libertarian freedom by asking another simple question:

> In some circumstance, do we possess the opportunity to exercise an ability to reject incoherent thoughts and beliefs in favor of coherent thoughts and beliefs?

Note the range of alternative options available from which to choose: incoherent thoughts/beliefs vs. coherent thoughts/beliefs.

Recall that if determinism is true and exhaustively describes the causal story of humanity, then if something else determines us to affirm a falsehood, then it is impossible for us to infer a better or true belief in the same circumstance. It is impossible for us to think otherwise if something or someone else is deterministically preventing us from thinking otherwise in that very circumstance.[16] If all things regarding a human are determined by something other than the human *qua* responsible agent, then there does not seem to be any room for epistemically responsible thinking. With that in mind, let us return to the last question. If we answer "yes", then our answer logically implies *libertarian freedom* as we affirm our opportunity to exercise an ability to choose between a range of alternative judgment options each of which is compatible with our nature and overall mental state at the moment of choice in a specific circumstance. However, if we answer "no", then several problems arise with respect to the notion of *reason* as illustrated in the following two questions:

1. Why trust our answer?
2. Why should anyone listen to our passive opinions about anything (including those on this topic)?

Further, is that which determined our thoughts and beliefs a reliable source? If not, then, significant problems arise.[17] If so, then how could we possibly know that?

Note that the implications of these problems arising from determinism are not merely the opinions of Christian theologians or apologists. Indeed, notable *atheistic* philosophers have also reached the implicit conclusion that is surfaced by the previous two questions: Libertarian freedom is necessary for the active use of reason. Moreover, the active use of reason is necessary for one to exemplify rationality itself, along with bearing rational responsibility for one's judgments, beliefs, etc.

## 3. Scholarly Support

Consider Evan Fales, for example. Fales is an atheist and philosopher who seems to support the case that libertarian freedom (specifically the categorical ability to choose or choose otherwise) is necessary to exercise the use of reason. Fales seems to be a libertarian who applies libertarian freedom to thinking, deliberation, and rationality. He says:

> I personally happen to have views about freedom of the will which are libertarian.

> I situate myself in a long tradition that tries most fundamentally to understand freedom of the will as a matter of having the capacity to exercise reason; rational choice. In fact, for me, the act of freely deciding is the act of rationally deliberating.

Unfortunately, Fales has it backward. It is the act of rational deliberation that is an example of acting freely, not the converse. Nevertheless, this problem does not affect his

overall point. Fales continues: "I'm a libertarian when it comes to free will, and that's not that common for atheists, or at least for naturalists".[18]

The reason, however, why most naturalists reject libertarian freedom is because libertarian freedom does not seem to fit nicely with most accounts of naturalism (this point will be addressed further below). John Searle (another atheistic philosopher) agrees that determinism is incompatible with rationality:

> Actions are rationally assessable if and only if the actions are free. The reason for the connection is this: *rationality must be able to make a difference*. Rationality is possible only where there is a genuine [libertarian] choice between various rational and irrational courses of action. If the act is completely determined then rationality can make no difference. It doesn't even come into play.[19]

Not only does Searle reject determinism, he clearly affirms that alternative possibilities are essential for rational action and choice earlier in the same book:

> Rationality is only possible where irrationality is possible. But the possibility of each requires freedom. So in order to behave rationally I can do so only if I am free to make any of a number of possible choices and have open the possibility of behaving irrationally

> When we perform conscious voluntary actions, we typically have a sense of alternative possibilities.[20]

Possessing access to alternative choice options (possibilities) in a specific circumstance is the epitome of strong libertarian freedom. Angus J. L. Menuge notes that Searle affirms a libertarian freedom that includes a genuine ability to choose otherwise: "Rationality presupposes an entity with libertarian free will that can act on some reasons rather than others". What Menuge means by "genuine ability" is categorical ability: given choices A or B, the agent can choose either without anything changing in the agent at the time of choice (Menuge 2013). This contrasts with the compatibilist, determinist notion of the ability to do otherwise. On this view, such ability is conditional. If certain beliefs/desires are present within the "agent", he necessarily must choose A. However, the "agent" could have chosen B if an alternative set of beliefs/desire had been present and necessitated the "choice" of B. This does not seem like "an ability to do otherwise" worth having.

To see why we say this, note first of all that our FTA provides defeaters for hard determinism and compatibilism construed as entailing determinism.[21] Furthermore, it is determinism that is the main problem addressed by our argument. This becomes evident when we focus our attention on the nature of deterministic, efficient-causal chains of events. The causal relation among the successive *relata* in such chains—the events—is transitive, and therein lies the rub. Suppose we take an arbitrary section of such a chain, say, events $e_3$ through to $e_9$. Given transitivity, if $e_3$ causes $e_4$ and $e_4$ causes $e_5$, then it is really $e_3$ that causes $e_5$ or better, $e_3$ causes $e_4$-causing-$e_5$. Here, $e_3$ is the actual efficient cause of $e_5$, and $e_4$ is merely a passive instrumental cause through which $e_3$ causes $e_5$.

However, recall that $e_3$-$e_9$ is an arbitrary section of a longer causal chain. Given transitivity, all the *relata* in the deterministic efficient-causal series are passive instrumental "causes"; i.e., caused causes. As has been shown repeatedly in the context of arguments for God's existence, without a First Unmoved Mover, and Uncaused Causer, no causation can be passed down through the passive chain of instrumental causes. Furthermore, such a First Mover is a libertarian agent with the power to initiate spontaneously causal action without having something else cause it to cause. This same argument applies to the need for a libertarian agent to initiate a series of instrumental causes in light of some teleological end. It is precisely the lack of any grounding for the originative source and subsequent transfer of efficient causal power that renders deterministic chains otiose. Admittedly, these points do not exhaust our development of the FTA, but they are an important part of it, and they are especially relevant to the claim that this argument succeeds against hard determinism and standard compatibilism.[22]

Returning to the issue of alternative possibilities, if Searle's notion of alternative possibilities is correct, then even if one asserts that libertarian freedom is possessed by humanity but the opportunity to exercise an ability to think or judge otherwise does not, then nothing about humanity ever really *makes a difference*. This is the case because no one would ever possess the opportunity to exercise an ability to think other than the way they actually do. In a world in which we all hold false beliefs, the power, as Searle says, to "make a difference" seems to be a power worth wanting. Lacking such implies that no one is rationally responsible for his or her beliefs.

Regarding this view, philosophers Moreland and Craig write: "If one is to have justified beliefs, then one must be free to obey or disobey epistemic rules. Otherwise, one could not be held responsible for his intellectual behavior".[23] The phrase "to obey or disobey" implies alternative possibilities when it comes to thinking, rationality, and "intellectual behavior". Accordingly, if one is not intellectually or epistemically responsible for their mental behavior, then one does not possess justification for any resulting belief. This is a major problem if justification is required for knowledge.[24]

The salient point is this: If rationally intuited or inferred knowledge claims are illusory, then one cannot rationally affirm such claims. However, the ability and power to rationally intuit, infer, and justifiably affirm knowledge claims seems to be desirable. We should not be too quick to give it up.

The well-known naturalist, Sam Harris, however, surrenders it rather quickly. Harris writes:

> Thoughts and intentions emerge from background causes of which we are unaware and over which we exert no conscious control.

> Either our wills are determined by prior causes and we are not responsible for them, or they are the product of chance and we are not responsible for them.

Accordingly, all of our thoughts, what we think *of*, how we think *about* it (intentionality), and ultimately all of our beliefs are either causally determined by antecedent conditions, or they are merely the product of chance, none of which is up to humanity. It follows, however, that if one assumes the exhaustive causal determinism of humanity, as Sam Harris does, then it is self-defeating to argue for the exhaustive causal determinism of humanity. Consider the fact that if the forces and events of nature causally determine Harris to affirm a false belief about X (in the actual world), then Harris does not possess the opportunity to exercise an ability to infer a better or true belief about X in that circumstance.

Next, consider the following argument:

A1    If exhaustive naturalistic determinism (hereafter, END) is true, then the forces and events of nature causally determine all humans—including Harris—to infer and affirm some false metaphysical beliefs (no one is infallible).

A2    If the forces and events of nature causally determine all humans to infer and affirm some false metaphysical beliefs, then Harris stands in no epistemic position to know which of his inferred metaphysical beliefs are true and which are false.

A3    If Harris does not stand in a position to know which of his inferred metaphysical beliefs are true and which are false, then Harris possesses a defeater against (a reason to doubt) said metaphysical beliefs.

A4    If Harris possesses reason to doubt said metaphysical beliefs, then Harris cannot rationally affirm said metaphysical beliefs (this would include his affirmations of END—that the forces and events of nature causally determine all things and that humans do not possess libertarian freedom to think).

A5    Therefore, if END is true, Harris cannot rationally affirm END (and likewise regarding any of his other inferred metaphysical beliefs).

Note that "the forces and events of nature" are outside the genuine control of the person. Diachronically, these forces and events are constituents in causal chains that began before the person was born—specifically, before the person began to reason and infer. Synchronically, these forces and events are constituents in causal chains that come from

outside the person, passively enter and pass through the physical and mental states of the person, and eventually cause passive body movements on their way out of the person. In this way, the person is a mere theater of physical and mental states that constitute a puppet show in the theater. Neither the theater nor the puppets actually *do* anything and, thus, are *not* rationally responsible agents.[25]

Recall Searle once again: "If the act is completely determined then rationality can make no difference".[26] Ultimately, it is simply self-defeating to argue for or affirm determinism. Epistemologist Robert Lockie concludes that if one affirms determinism, then one "cannot be epistemically justified in her embrace (adoption, articulation, and defence) of determinism". Epistemic justification seems to be worth desiring and having, whereas self-defeat is not worth desiring (Lockie 2018).

Graham Oppy (2022) wrote that "there is next to nothing in metaphysics on which there is even bare majority agreement among metaphysicians". Indeed, the same can be said of Christian theologians discussing anything besides the proposition that God raised Jesus from the dead—what C. S. Lewis ([1980] 2009) referred to as *mere Christianity*. It stands to reason that nature or God has not granted metaphysicians or theologians with infallibility or a perfect set of metaphysical or theological beliefs.[27] Since it seems that something or someone else (like God) is not determining humanity to always reach true beliefs, then it follows that if one's noetic structure is functioning properly, then libertarian freedom is (at least in some circumstances) a vital ingredient in the rationality mix. Hence, if the broad ability to be a free thinker has been lost—and one is not free (or does not have opportunities) to actively employ the use of reason in an appropriate circumstance, then something seems to have gone terribly wrong with one's thinking faculties (perhaps they have been damaged via brain tumor, head trauma, or one has failed to take one's thoughts captive for far too long and caused self-inflicted damage to their own cognitive faculties). Consider this a kind of *epistemic evil*.

Since we are obviously not the kind of creatures who are always determined to believe truth, we better hope that we are not determined to affirm false metaphysical and theological beliefs. Indeed, it seems that if EDD is true, then we are left with nothing but skepticism. Surely, the power to infer true beliefs is greater and more desirable than being passively determined to affirm false beliefs. In a world suffused with false beliefs, the active use of reason to rationally infer better and best (let alone true) explanations, and rationally affirm knowledge claims—*something definitely worth wanting*—requires libertarian freedom.

## 4. A Mad Scientist to the Rescue

The following thought experiment illustrates the necessity of libertarian freedom when it comes to rationality and the active use of reason. Suppose a mad scientist has somehow implanted microchips in a person's brain while she was sleeping and now exhaustively controls and determines all of her thoughts and beliefs all the time. This includes exactly what she thinks of and about and, more importantly, exactly how she thinks of and about it (i.e., all of her evaluations and judgments).

All of her thoughts about her beliefs and all of her beliefs about her thoughts are now determined by the mad scientist (she has no opportunity to think or believe otherwise). She has zero "guidance control" over what she thinks of or how she thinks about it (all of her passive thoughts are completely up to and at the mercy of the whims of the mad scientist).[28] She does not have the control condition required for rational responsibility. In other words, she does not meet the "control condition" if something or someone else is in complete control of her condition.

That is to say, if something (non-rational) or someone else (who is untrustworthy) controls all of her reasoning, then there is reason to doubt her reasoning. If something or someone else controls how she reasons, then there is reason to doubt her "use of reason". Indeed, she would lack the kind of libertarian agency to actively use her reason. Instead, her succession of thoughts would not be hers at all but, rather, those of her controller. If

she is not a libertarian free thinker, then something or someone else is ultimately in control over all of her thoughts, intuitions, beliefs, evaluations, and judgments.[29]

She also does not have control over the next words that will form in her mind or come out of her mouth. If all things are exhaustively determined, then she has no active control over any of her thoughts (which seems to be antithetical to the teachings of both Jesus and Paul).[30] What is more, or rather, much less, she is nothing but a "passive cog" as epistemologist John DePoe has indicated: "Ultimately [on determinism] the human agent is downgraded from being a person with active powers of rationality to a passive cog that is at the mercy of causes beyond one's control" (DePoe 2020).

Not only does this passive cog-ness seem to oppose the teachings of Scripture, it also stands opposed to our direct experience. Speaking of our direct experience of the libertarian freedom to actively think and "take thoughts captive" as the apostle Paul would say, Moreland observes the following:

> Most of us are quite aware of the different what-it-is-like, the different phenomenological texture in having a passive thought (e.g., when someone is talking to me) and an active thought (one to which I exercise my active power and choose to attend). This difference is self-evident to introspective awareness. And such awarenesses provide nondoxastic, internalist grounds for the proper basicality of one's belief that one has and can exercise active power. On this basis, I think that active power is epistemically, conceptually, and ontologically foundational and essential for there to be such things as knowledge of, a concept of, and the existence of libertarian acts. And it is on the basis of this account, and knowledge of relevant moral features that we have the concept of moral responsibility.

Sure, a passive cog might possess *passive awareness,* but one would never have active control of what one is aware on this deterministic view. One would not possess the ability or the power to "be more careful" while thinking things through. One's level of so-called "carefulness" is also causally determined by the nefarious neurosurgeon.

If determinism is true, then when one passively experiences sensations[31] of deliberation about what one ought to believe, the fact of the matter is that every other belief option—besides what God, nature, or a mad scientist causally determined the person to believe—was closed off and locked away from the person's access. It might have subjectively *seemed* as if the person experiencing these sensations of deliberation could have accessed these other options, but objectively speaking, it was an illusion (also causally determined by something or someone else).

The ability to actively choose to be "more careful" while thinking things through seems to be something worth wanting. With this in mind, if one is still hanging on to determinism, another question arises:

> How can she (not the mad scientist) rationally affirm her current beliefs as good, bad, better, the best, worse, the worst, true, false, probably true, or probably false without begging the question?

This seems to be an impossible task. Epistemologist Jim Slagle makes the point:

> According to determinism, however, the only methods by which we could examine such processes are products of these processes themselves. To appeal to the reliability of these processes in arguing for their reliability is an invalid procedure. This may be able to show that the processes in question are *not* reliable (by showing that they lead to an incoherent system, for example), but to appeal to these processes in order to verify the reliability of these processes is simply, and blatantly, to beg the question.[32]

No matter how she responds, the response is not from her (as a source), but ultimately determined by and up to the mad scientist. That is to say, unless she has libertarian freedom, the response will not be from her; it simply passes through her. If we replace the mad scientist with "physics and chemistry", "God", or anything else, then we have the exact same rationality problems but for different reasons. As epistemologist Kelly Fitzsimmons

Burton aptly notes: "Proper function of our cognitive faculties must first rule out the [deterministic] influences of any outsiders such as Alpha Centurion cognitive scientists, Cartesian evil demons, and also internal influences such as a brain lesion or even the influence of mind-altering substances. All of these influences may cause one's faculties to fail to function properly".[33] We may add "a deity of deception" to that list.[34]

Jim Slagle writes:

> The claim, recall, is that there must be an *explanation* for a belief, it must be a *good* explanation, and it must be *my* explanation. But if the determining factors are extrinsic to the individual (not to the belief), then it is difficult to see how it could [be] *my* explanation as opposed to just *an* explanation. In order for it to be my explanation, I have to accept it. If my acceptance is also determined by extrinsic forces [which could include God], then in what sense is it my explanation? In fact, how could the resulting state be called a "belief" at all? Belief seems to involve both reception of information *and some level of assent to or approval of it*. After all, we often receive information that we do not subsequently believe, so clearly mere reception of information is an insufficient definition of belief. And assent or approval in turn seems to intrinsically involve the concept of *self-origination*. I may not have to originate the explanation of the belief, but I do have to make whatever explanation there is my explanation for believing it. This would suggest that determinism is incompatible with belief, and so belief in determinism, including theistic determinism, is self-defeating.[35]

Self-defeat is the last thing a rational person wants. Sure, the mad scientist *could* causally determine her to hold some true beliefs, but *she* would never be in a position to judge, evaluate, or *know* if what the mad scientist is causing her to believe is correct (since her judgments and evaluations are also determined by the same mad scientist). This is why it does not matter if the external controller is a good, trustworthy being (e.g., God). Given determinism, a human is not really a deliberating agent. As a result, that human has no real ability to reason through syllogisms or engage in any intellectually responsible mental act. Thus, with God as the deterministic controller, it may be the case that one's thoughts, beliefs and so forth are true, but no one would ever be able to know that by reasoning.

Moreover, on determinism, we stand in no position to judge or evaluate what false beliefs this nefarious neuroscientist is imposing upon us while also determining us to think those false beliefs are good, true, or probably true. We would have no ability to choose to accept or reject information. Our so-called acceptance or rejection of new information is also not up to us but determined by the mad scientist (either directly or via how he programmed and determined our cognitive belief-forming faculties to operate).[36] Remember, we are discussing *exhaustive* determination. Everything that happens during our so-called "reasoning process" happens exactly the way the mad scientist desires and determines—even when "we" reason poorly. To make matters worse, this mad scientist even makes some people believe that he is infallible so that they can trust the beliefs he imposes upon them. For that matter, he could make people believe that he has all of the divine attributes.[37]

Sure, in a denuded sense, reasoning plays a role on a deterministic view. However, the problem is that non-rational forces or untrustworthy deities causally determine the exact manner in which one reasons or in what manner one responds to reasons. After all, one can reason poorly, reach horrible conclusions, and happily affirm false beliefs—ultimately based upon the non-rational forces of nature or untrustworthy deities.

Thus, according to deterministic views, we seem to have undercutting defeaters to many important beliefs. If the mad scientist controls and determines our entire thought life (directly or indirectly), then we would not be in a position to judge for ourselves whether we ought or ought not to accept information and believe it, or reject information and not believe it, as the Apostle Paul challenges in 1 Corinthians 10:15. The ability worth

wanting—to accept or reject information—is not under *our* control but rather the control of the mad scientist.

Indeed, when one subjectively feels they are evaluating or judging a currently affirmed belief (to see if it *should* be affirmed as true or false), in reality, the same nefarious neuroscientist who causally determined the affirmation of the false belief in the first place is also the same mad scientist who is causally determining the so-called "evaluation" of the affirmed false belief. This would be akin to "fact checkers" assuring us that a particular cable news channel is reliable (and not "fake news"), but then we find out that the so-called fact checkers are employed by the same cable news channel. If that is the case, no epistemic progress has been made. Epistemic progress, however, is valuable.[38]

We think that these arguments are persuasive. However, we also recognize that compatibilists usually retain their determinism in light of them. In our view, the most sophisticated defenses of the consistency between determinism and rational deliberation and responsibility have been proffered by Derk Pereboom. Pereboom's case is frequently cited as a defeater for the arguments we have presented above. Unfortunately, space considerations prevent us from assessing his views with the detail they deserve. However, we would be remiss if we did not provide a brief examination of them.

According to Pereboom, given determinism, there are two epistemic conditions an agent must satisfy to deliberate about which of a number of alternatives the deliberator may choose. To engage in deliberating, one must believe in both of them:

- (S) An epistemic openness condition: To deliberate rationally among distinct actions $A_1 \ldots A_n$, subject S cannot be certain of the proposition that she will do $A_i$, and either (a) the proposition that S will do $A_i$ is consistent with every proposition that, in the present context, is settled for S (S believes it and disregards any doubts about it when deliberating), or (b) if it is inconsistent with some such proposition, S cannot believe that it is.

- (DE) A belief in the efficacy of deliberation condition: In order to rationally deliberate [*sic*] about whether to do $A_1$ or $A_2$, where $A_1$ and $A_2$ are distinct actions, an agent must believe that if as a result of her deliberating about whether to do $A_1$ or $A_2$ she were to judge that it would be best to do $A_1$, then, under normal conditions, she would also, on the basis of this deliberation, do $A_1$; and similarly for $A_2$.

Pereboom holds that satisfaction of these two conditions is sufficient for epistemically responsible deliberation even though determinism is the case. Further, he holds that someone who *believes* in determinism and also believes S and DE can deliberate without inconsistent beliefs. Has Pereboom successfully made his case? For several reasons, we do not think so.

For one thing, there are two serious problems with (S). First, (S) fails because it focuses on the wrong issue, one that is derivative and not fundamental: (S) erroneously makes the issue one of alternative possibilities rather than focusing on the sort of control required for real deliberation grounded on the nature of deliberation itself. Below we will unpack that nature, but presently, we note that this error shifts the focus away from ontology to epistemology. This shift simply changes the subject and muddies the waters. Real deliberation entails that $A_1 \ldots A_n$ must actually be ontological possibilities, especially when we note that deliberation is an internal mental process that does not have a bodily "action" among its constituents. Thus, Frankfurt-style counterexamples are both irrelevant and misleading since the person in those cases retains the active power to do something or to refrain from doing something in which case the observer causes the relevant body movement. Thus, the person retains active power and alternative possibilities. Unfortunately, (S) only gives us *ersatz* deliberation, not genuine deliberation, because (S) does not include active power or genuine, ontological alternatives.

Relatedly, the person in (S) has a false belief, viz., that $A_1 \ldots A_n$ are actually real, genuine possibilities when they are not. Given determinism, only one alternative must obtain. Because (S) requires that the person is shielded from knowing that alternative, the

"deliberator" is actually deluded and operating under pretense. If a philosophical solution to some problem requires such a thing, that is sufficient grounds for rejecting that solution.

So much for (S). What about (DE)? It is also a failure and by seeing why, we get to the heart of the fundamental, basic issue about what is wrong with deterministic accounts of deliberation, including (DE). Put simply, the person in (DE) has a false belief about the efficacy of her deliberating based on a false belief about the nature of deliberation itself. To see this, let us begin by asking how it is that we came up with the notion that we engage in deliberation. From whence commeth this notion in the first place? How do we know that we actually deliberate?

We think there is an obvious answer to these questions. Chronologically and epistemically, one begins by having first-person direct awareness/access or knowledge by acquaintance with one's own conscious states and their flow. When one engages in a process of deliberation, one is simply aware of those conscious states that constitute that process, as well as those states before and after the process. Further, by attending to such a process, one becomes directly aware of certain essential characteristics of deliberation: (a) One exercises active power instead of merely undergoing the triggering of a passive liability. For example, there is a phenomenological difference between an active and a passive thought. (b) One is aware that she herself is the agent performing the act of deliberation. One is aware of not being a passive patient through which a deterministic chain of efficient causes is running through her, all the while bypassing the superfluous "agent" altogether. (c) One becomes aware that her deliberative process has active, intrinsic teleology. She originated the process, performs acts of weighing and weighting different factors, of choosing to attend to certain factors and refrain from others, all for the sake of reaching a conclusion that is true, or at least, most reasonable. (d) One becomes aware of cases in which she experiences genuine epistemic akrasia. She may become aware of where the process is going and choose to refrain from drawing the most rational conclusion by suppressing certain evidence of which she is aware, rationalizing about the weight of other pieces of evidence, and so forth.

On the basis of repeated cases of attending to the features of one's deliberative process, one forms a concept of genuine deliberation, and on that basis, will recognize that (DE) does not involve genuine deliberation.

Though it is not essential to our argument, there is another related issue worth mentioning. To see this, consider an example of the Knowledge Argument.[39] Suppose the world's leading expert on physics and the neurophysiology of seeing was born blind. She knows all the physical facts about seeing, but one day she miraculously gains the ability to see. She now learns a whole new set of facts, e.g., she becomes aware of the color red and of what-it-is-like to see red. Based on repeated experiences of red, she eventually forms the concept of being red. This concept was unavailable to her before the miracle. Why? Her concept was formed on the bases of repeated occasions of knowledge by acquaintance with redness itself. Thus, the simple fact that she has that concept entails that she has been directly aware of redness via acquaintance.

Now we suggest that we all have a concept of active power, passive liabilities, teleology, being a first-mover who originates action and so forth. If this is so, then as seems evident in the case of the blind neuroscientist, it would appear that we all have knowledge by acquaintance with the real nature of deliberation and the essential traits that constitute it.

Given our account of the role of direct access to one's deliberative process in forming an accurate concept of genuine deliberation, upon reflection, other things become evident. Genuine deliberation entails the presence of real ontological alternative possibilities for the conclusion drawn. (S) and (DE) are bereft of these essential features and, accordingly, yield *ersatz* deliberation and not the genuine article. The ability in genuine efficacy is the exercise of first-moving active power. Thus, if the person in (DE) believes in the efficacy of deliberation, she believes in libertarian freedom. If she does not believe in libertarian freedom, she does not believe in the genuine efficacy of deliberation. Either way, (DE) is in trouble.

It is worth noting that when determinists make their case for epistemically responsible deliberation they employ certain terminology that is both unavailable to them and provides the real intuition-pump for the adequacy of their case. The terminology in view includes "agent", "acting", "deliberating", "efficacy", and so forth. Observe that all of these are typically understood as exercises of active power and not the mere passive triggering of passive liabilities as events in a series of passive (triggered triggers) occurrences running through, yet not involving the agency of the person. These terms are in the active and not the passive voice. For each employment of one of these terms, a reductive paraphrase is available that provides an accurate account of what is really being expressed. Consider the following:

(P$_1$) Rae deliberated about P.

(P$_2$) A deliberating-about-P was caused to occur in Rae.

P$_1$ employs "deliberated" in the active voice and implies that Rae was exercising her agency and active power in performing a deliberative act. However, this is not the case if determinism is true. For the determinist, P$_2$ makes clear what is actually meant by asserting P$_1$. Though unintentional, by utilizing statements like P$_1$ in expressing their views, determinists benefit from a rhetorical mirage that keeps hidden what is actually being asserted (P$_2$).

For these reasons, we judge Pereboom's case to be a failure.

## 5. The Free-Thinking Argument

Thus far, we have argued that there are some kinds of rationality and knowledge that are worth wanting and worth having and these kinds of rationality and knowledge (namely, the ability to rationally infer and affirm claims of knowledge) are not compatible with exhaustive causal determinism—where something other than us determines everything about us. This can be summarized in a syllogism:[40]

B1　If humanity does not possess the libertarian freedom to think, then humanity is never epistemically responsible.

B2　Humanity is occasionally epistemically responsible.[41]

B3　Therefore, humanity possesses the libertarian freedom to think.

Note that this argument says nothing directly about naturalism, atheism, materialism, physicalism, Calvinism, Molinism, or any other "-ism" that might be relevant to this discussion. However, speaking of "isms", this argument does conclude that humanity possesses the libertarian freedom to think. If that is true, then we have indirectly demonstrated that incompatibilism is true. That is to say, epistemic responsibility is not compatible with determinism.[42]

Moreover, this argument says nothing about alternative possibilities. This is simply something that all people who consider themselves to be rational ought to affirm. That is, if one is epistemically responsible, then one is a free thinker in a libertarian sense.

The fact is that if we do not possess libertarian freedom, then something non-rational or someone untrustworthy (like a mad scientist) is causally determining all of our thoughts and beliefs. If something non-rational or someone untrustworthy causally determines a person to affirm a false belief in a certain circumstance, then he or she has no ability to infer a better or true belief in that same circumstance. The view that non-rational forces ultimately determine all of our thoughts and beliefs is a view that cannot be rationally accepted any more than the belief that an untrustworthy mad scientist determines all of our thoughts and beliefs should be trusted.

Suppose, for example, that God determines Pastor Jones to study Scripture for decades and finally conclude that the eschatological position known as the premillennial view is true. Unbeknownst to Pastor Jones, however, in actuality, the post-millennium view is true. If that is the case—and God determines Jones to think the premillennial view is true until the day he dies—then Jones cannot do anything to infer a better or true eschatological belief and he will teach his entire congregation this false theological belief.

On this deterministic view, God is untrustworthy and Jones cannot infer better or true eschatological beliefs (although Jones confidently declares he has rationally reached his conclusion and that he is, in fact, proclaiming the truth regarding the end times).[43] Replace God with "non-rational forces", "a mad-scientist", or any other deterministic option, and we have epistemic problems. If one has no idea which of his or her beliefs are false, then he or she cannot rationally affirm which beliefs are true. This ability to infer better and true beliefs seems to be a power worth wanting, and, more importantly, one with which we have an intimate introspective awareness. The ability to rationally affirm claims of knowledge also seems to be a power worth wanting. In fact, that is a power we have. If one disagrees, one seems to assume that he or she has that power, too.[44]

Now, libertarian freedom—especially the libertarian freedom to think—is something all rational people *should* affirm. We are comforted by the fact that there seems to be a growing number of both atheists and Christians who are affirming libertarian freedom. There even seems to be a growing number of Calvinists who are rejecting EDD and affirming limited libertarian freedom even while still maintaining that soteriological matters are completely determined.[45] The popular apologist Greg Koukl is a wonderful example. He writes:

> The problem with this view [i.e., determinism] is that without freedom, rationality would have no room to operate. Arguments would not matter, since no one would be able to base beliefs on adequate reasons. One could never judge between a good idea and a bad one. One would hold beliefs only because he had been predetermined to do so.

> Although it is theoretically possible that determinism is true no one could ever know it if it were. Every one of our thoughts, dispositions, and opinions would have been decided for us by factors out of our control (Koukl 2019).

> Arguments for determinism self-destruct.

With that said, what is the best explanation for this libertarian freedom possessed by humanity—this power that seems to separate us from the animals and everything else in the universe?

The syllogism above can be expanded by incorporating two additional premises. This leads to two more deductive conclusions along with one abductive conclusion. As Moreland notes, the proponent of robust naturalism[46] ought to express a

> scientistic version of philosophical monism according to which everything that exists or happens in the world is (in principle) susceptible to explanations by natural-scientific methods in the hard sciences. Whatever exists or happens in the world is natural in this sense. Prima facie, the most consistent way to understand naturalism in this regard is to see it as entailing some version of strong physicalism: everything that exists is fundamentally matter. No nonphysical entities exist, including emergent ones. This constitutes a strong sense of physicalism and robust naturalism.

> The Grand Story [i.e., the account of how all things whatever came to be] is deterministic in two senses: diachronically, such that the state of the universe at any time t coupled with the laws of nature determine or fix the chances for the state of the universe at subsequent times; synchronically, such that the features of and changes regarding macro-wholes are dependent on and determined by micro-physical phenomena.[47]

Peter van Inwagen, giving a hat-tip to the late Carl Sagan, describes naturalism as "the thesis that nature is all there is or was or ever will be".[48] Inwagen goes on to explain that "naturalism implies that everything that exists is part, large or small, of the physical universe, and that the laws of physics, the laws that govern the behavior of and the mutual interactions among the parts of the physical universe, apply universally and without exception to everything that exists".[49]

To put it simply, Alvin Plantinga has described naturalism as the view that "there is no such person as God or anything like him".[50]

If one affirms that God or things like God do or do not exist, the following question should be raised:

If God exists, what would God be like?

Once we know the answer to that question we must ask an additional question:

*What Are Things That Are Like God Like?*

Not only does the Bible tell us what God is like (e.g., "God is spirit", "God is love", etc.), but our understanding of God is also informed by natural theology. For example, from the Kalām cosmological argument, we know that God is an immaterial being who has active causal power. When we combine the fine-tuning argument with the Kalām, we know that God is an intelligent immaterial thinking thing[51] with active causal power.[52] Thus, by inference, if God has created anything like him,[53] it would be fair to say that it would be an immaterial thinking thing with active causal power. This immaterial thinking thing with active causal power is what we refer to as the supernatural soul created in the image of God. Indeed, in this sense, each and every human being is like God.[54]

With this in mind, consider one particular version of the FTA against naturalism:

C1    If robust naturalism is true, God or things like God do not exist.
C2    If God or things like God do not exist, humanity does not freely think in the libertarian sense.
C3    If humanity does not freely think in the libertarian sense, then humanity is never epistemically responsible.
C4    Humanity is occasionally epistemically responsible.
C5    Therefore, humanity freely thinks in the libertarian sense.
C6    Therefore, God or things like God exist.
C7    Therefore, robust naturalism is false.
C8    The best explanation of God, things like God, and the libertarian freedom of humanity is the biblical account of reality.

The final conclusion, unlike the previous three deductive conclusions, is an abductive move and the beginning of a new conversation.[55] As Moreland explains: "The argument for God's existence from the reality of robust libertarian freedom seeks to show that, given robust freedom, a theistic versus a natural-scientific explanation is epistemically and explanatorily superior".[56]

More fully:

> Applied to our case, the claim is made that, on a theistic metaphysic, one already has an instance of an unembodied mind in God which exercises robust libertarian freedom. Therefore, it is hardly surprising that embodied or unembodied finite libertarian agents should exist in the world. But on a naturalist view, mental entities are so strange and out of place that their existence (or regular correlation with physical entities) defies adequate explanation. There appear to be two realms operating in causal harmony and theism provides the best explanation of this fact.[57]

While we believe that a biblical view of God is the best explanation of human libertarian freedom and the soul, for the sake of time and space, we will simply defend each premise leading to each deductive conclusion in this paper. The first three steps of the argument are rather straightforward. To succinctly defend each premise, C1 is true by definition: If naturalism is true, nature (as described above) is all that exists. Thus, things other than nature (like God and things like God)—what philosophers mean by *supernatural*—would not exist.[58]

So, the first premise is non-controversial. Let us examine the next (C2):

> If God or things like God do not exist, humanity does not freely think in the libertarian sense.

While Christian physicalists might not like the second premise, this new wording allows them to join the party. Over the past ten years, I (Stratton) have used the second premise to bring immaterial and supernatural souls into the discussion. While we still believe substance dualism[59] is the best way to think about these things, Christian physicalists can affirm the new and updated wording of this second premise. Admittedly, postulating a purely physical thing possessing the powers of libertarian freedom to think and infer best explanations regarding metaphysical and theological questions seems to be far less plausible than the existence of an immaterial thinking thing with a body that possesses those powers. However, setting that aside, for present purposes we grant that a Christian physicalist could embrace the former option and thereby have no problem with accepting premise C2.

Christian physicalists reject naturalism because they believe in a supernatural, immaterial, and non-physical Creator of the physical universe. So, they still have a good reason to affirm the fact that humans are miraculously endowed with the power to think freely in the libertarian sense. Moreover, although they do not think we are "*like* God" in the sense of being an immaterial free-thinking thing, they can affirm that humanity is like God in the sense that we possess the supernatural power of libertarian free thinking.[60]

Now that Christian physicalists can affirm the second premise, let us discuss the naturalist's view of reality. Premise C2 is tantamount to the following:

> *If all that exists is nature, then everything about humanity—including all of our thoughts and beliefs—are determined by the forces of nature, the initial conditions of the big bang, and perhaps some indeterministic quantum events (all of which are outside of human control).*

Indeed, it stands to reason that probably, if we are purely physical kinds of things, then everything about us would ultimately be determined by the non-rational laws and events of nature.

Of course, some might hypothesize that a supernatural soul or immaterial thinking thing might somehow evolve or *emerge* from purely physical stuff and then somehow turn around and control the physical body.[61] Not only does this seem ad hoc, but it also seems to us that this view does not seem to count as official naturalism, since a thing "like God" exists even if God, on this view, does not exist. Moreover, it would violate the view advanced by Sagan (and endorsed by Van Inwagen), that naturalism is the view that purely physical objects are "all that ever will be". In order for an object to be purely physical, all of its properties and parts must be purely physical.

It is hard to take this view seriously. Many views are strictly *possible*, but this view seems drastically unlikely, ad hoc, and even miraculous. At the least, it is fair to say that if God does not exist, then immaterial, rational, thinking things, with active causal power and libertarian freedom *probably* do not exist either.

The naturalist John Bishop (1989) is clear: "Agent-causal relations do not belong to the ontology of the natural perspective. Naturalism does not essentially employ the concept of a causal relation whose first member is in the category of person or agent (or even, for that matter, in the broader category of continuant or 'substance'). All natural causal relations have first members in the category of event or state of affairs".

All that to say, we have good reason to affirm premise C2. What about the third premise?

> *If humanity does not freely think in the libertarian sense, then humanity is never epistemically responsible.*

Premise C3 is virtually equivalent with the following statement:

> *If something or someone else determines person P's thoughts and beliefs, P is not responsible for her thoughts and beliefs.*

It expresses the fact that if all things are causally determined, then that includes all thoughts, intuitions, beliefs, evaluations, and judgments. If all of a person's thoughts, intuitions, beliefs, evaluations, and judgments are always forced upon her, then she is simply left assuming that her determined thoughts, beliefs, evaluations, and judgments are good and that her beliefs are true.[62]

Therefore, one could never rationally affirm that her beliefs really are the inference to the best explanation. Given all that has been said, this belief can only be assumed without warrant or justification. Moreover, this question-begging assumption is another instance of passivity as it would likewise be causally determined and forced upon her by something or someone else. Accordingly, she could not do anything other than assume, and assume exactly as she has been determined to assume.

If a belief is merely assumed it is not a justified belief (especially if one assumes belief is determined by something non-rational or someone who is untrustworthy). What is justified on this deterministic view, however, is a sense of vertigo.[63] Vertigo is not worth wanting. Justified true beliefs, however, are something we should all strive to attain. They are definitely worth wanting and are not illusory. Thus, premise C4 is something that we must accept as true, because to rationally argue against it affirms it.

The FTA aims to show that if something non-rational or someone who is untrustworthy determines all of our thoughts and beliefs—our entire reasoning process—then we have reason to doubt our inferred thoughts and beliefs based upon "our" reasoning process. It makes much more sense to affirm that God exists and created us with cognitive faculties with the powers of free thinking in a libertarian sense, the opportunity to think carefully, take bad thinking captive (2 Cor 10:5), and infer true metaphysical and theological beliefs over time.

The naturalist—who believes that God and things like God do not exist—typically affirms, as they should, that some non-rational cause determines all effects regarding humanity. This would include, however, the effect of every one of their thoughts and beliefs. This is a serious problem.

Perhaps the one who affirms that God does not exist might want to affirm human libertarian free thinking without explaining how they have it. Indeed, as pointed out above, some prominent atheist philosophers do affirm the libertarian freedom of humanity. They are closer to truth. However, if God did not design and create us to be rational free thinkers, it sure seems like all thoughts and beliefs on this indeterministic view would merely be random as opposed to rational. Libertarian and rational free thinking—designed to gain justified true beliefs (if we are careful)—makes better sense on a theistic view of ultimate reality.[64]

Moreland notes the following:

> It should be clear that the appearance of libertarian agents and free will is natural in this theistic view but quite odd, unnatural, and not basic in a naturalist worldview. Thus, again, it may very well be ad hoc and question-begging for someone to claim that a view of libertarian free will is "at home", "consistent with", or "not ruled out by" naturalism. And if such a view is, in fact, developed, it may either be so minimalist that it either leaves out important features of a libertarian account and, thus, may need to prove its libertarian credentials, or so irrelevant to plausible versions of libertarianism that match our experiences of agency, that its "consistency" with naturalism may involve a view that can be safely ignored.[65]

We are skeptical that some minimalist version of both naturalism and libertarian freedom are compossible, at least a version that some naturalists would accept. Moreland elaborates on this by adding that "the proper question is not, 'Can a minimalist version of naturalism and libertarianism be shown to be logically consistent?' Rather, it is 'Given the most reasonable form of naturalism and theism as a rival worldview, is it more reasonable than not to believe that the existence of libertarian actions and agents are more at home in a naturalist worldview than a theistic worldview? What is the truth of the matter?' "[66]

God creating humanity in his image and likeness with the power to be rational—and approximate to his perfect standard of knowledge—or not, provides a much better explanation than a robust naturalistic story of the world. With that in mind, libertarian freedom seems to be evidence pointing to the existence of God. That is a win for theism.

Unfortunately, however, not all Christians are able to rejoice in this victory. Divine determinists (many if not most Calvinists) also find themselves in the same sinking ship of problems highlighted throughout this discussion. They have the same problems but for different reasons, since God, on their view, determines all thoughts and beliefs. Consider the words of notable Calvinist, Matthew J. Hart: "Calvinists are theological determinists. They hold that God causes every contingent event, either directly or indirectly".[67] Since human thoughts and states of belief are contingent events, this means that God, according to Calvinistic determinism, causes each and every thought and belief, including all of our false and evil beliefs.

In his work titled *The Providence of God*, Paul Helm explains where our thoughts come from according to his Calvinistic view: "Not only is every atom and molecule, every thought and desire, kept in being by God, but every twist and turn of each of these is under the direct control of God. He has not, as far as we know, delegated that control to anyone else".

Helm is incorrect. Scripture is clear that we do have some control of our thoughts and ultimately some of our important beliefs. After all, the apostle Paul declares that we are the kind of creature who can take certain thoughts captive (2 Cor 10:5) before they take us (Col 2:8). Thus, these thoughts Paul refers to cannot be necessitated by antecedent conditions which are outside of our control. The committed Calvinist/determinist might counter and say he does not have the same problems as the naturalistic determinist since all of his thoughts and beliefs are determined by an omniscient God of truth who is the standard of perfect knowledge.

Unfortunately, the problems are much worse for the divine determinist. Unless the divine determinist is going to claim to be theologically infallible, this move is not going to work. That is, since no one is theologically infallible, it follows that every person affirms false theological beliefs. If that is the case it follows that God determines all people—including all theists, which includes all Calvinists—to affirm false theological beliefs.[68] This would mean that our Creator is *not* a God of truth who desires all people to know the truth (1 Tim 2:4) and, instead, our Creator is relegated to a "deity of deception".[69]

A deity of deception—a god who both desires and causally determines all humans (including all of the deity's loyal followers) to affirm false beliefs—is not worth worshipping. If this is the case, and a deity of deception determines all of our beliefs, then we have reason to doubt our beliefs, including our theological beliefs. This can be clarified by the following syllogism:[70]

D1   For any Christian c, there is at least one theological belief b such that b is false and c affirms b.[71]

D2   If EDD is true, then, for any Christian c and any theological belief b, if c affirms b, then God determines c to affirm b.

D3   Therefore, if EDD is true, then, for any Christian c, there is at least one theological belief b such that b is false, c affirms b, and God determines c to affirm b.

Clearly, the deductive conclusion implies that, if EDD is true, then God determines all Christ followers to affirm at least one false theological belief (and based upon the "fake news" analogy above, we stand in no position to know which of our determined theological beliefs are false). Thus, if EDD is true, not only do humans lose the power to rationally infer and affirm truth, it seems that the deity of EDD is anything *but* the God of truth. If God is merely a deity of deception (something far less than a maximally great being), then God does not seem to be significantly different from a Cartesian demon. If a Cartesian demon or a deity of deception causes and determines all of our thoughts and beliefs, then we have reason to doubt important thoughts and beliefs. To reiterate what Craig and Moreland said:

"If one is to have justified beliefs, then one must be free to obey or disobey epistemic rules. Otherwise, one could not be held responsible for his intellectual behavior".[72] If one is not responsible for his intellectual behavior, then determinists should never condemn, scorn, blame, or be frustrated with one who disagrees with them.

Recall Slagle's words once again: "If my acceptance [of a reason to believe X] is also determined by extrinsic forces [which could include God], then in what sense is it my explanation? In fact, how could the resulting state be called a "belief" at all? Belief seems to involve both reception of information *and some level of assent to or approval of it*".[73]

However, if determinism is true, one's level of assent or approval of information is determined by something or someone else. If that something or someone else is untrustworthy (just as a demon, a god of mischief, or a deity of deception would be), then undercutting defeaters abound. Accordingly, justification and rational affirmation devolve into question-begging assumptions. If that is the case, then knowledge is illusory. Epistemologist John DePoe agrees: "Given that the nature of justification requires some sense of freedom or autonomy, exhaustive divine determinism is incompatible with humans possessing knowledge".[74]

With justified beliefs in mind an additional argument arises:

E1   If EDD is true, then God determines all Christians to affirm some false theological beliefs.

E2   If God determines all Christians to affirm some false theological beliefs, then God is deceptive and His Word (the Bible) cannot be trusted.[75]

E3   God is not deceptive and His Word can be trusted.

E4   Therefore, God does not determine all Christians to affirm some false theological beliefs.

E5   Therefore, EDD is false.

This deity of deception argument says nothing about humanity possessing alterative possibilities.[76] It merely shows that EDD is false and that humans must be the proper source of at least some of our thoughts and beliefs. If one assumes that something or someone else determines all of his thoughts and beliefs, then one cannot rationally affirm important thoughts and beliefs. But of course, at least on occasion, we can rationally affirm these thoughts and beliefs (to argue otherwise presupposes it). Therefore, something or someone else does not determine all of our thoughts and beliefs. Therefore, we are libertarian free thinkers.

## 6. Conclusions

Ultimately, a person's metaphysical and theological beliefs are either: (i) determined by something non-rational (and thus, untrustworthy), (ii) determined by a deity of deception (and thus, untrustworthy), (iii) random (and thus, untrustworthy),[77] or (iv) caused by an intelligently designed[78] free-thinking agent created in the likeness of a maximally great being (God) with cognitive faculties functioning properly (subject to no dysfunction) in an appropriate environment which can be aimed at truth if the agent is careful and handles his or her powers responsibly.[79] The first three options leave us with skepticism and reason to doubt our metaphysical and theological thoughts and beliefs.[80] Option (iv) is the best explanation and our best hope.[81] However, the fourth option entails that one is free in a libertarian sense—not determined by something unreliable or someone who is untrustworthy.

If one believes that he or she is a rational free-thinker who is not ultimately mind-controlled by something (or someone) else, then one should reject the determinism that seems to follow from both naturalism and EDD. Instead, one ought to affirm that a supernatural God exists. Moreover, one ought to realize that he or she is a supernatural and immaterial active and rational free-thinking thing—a soul—created in God's image and likeness, and who will survive the death of one's physical body.[82] Arguing for this claim

must be left for another occasion. Nevertheless, we believe the FTA is ultimately sound and opens the door to multiple important metaphysical realities.

**Author Contributions:** Writing—original draft preparation, T.A.S. and J.P.M. All authors have read and agreed to the published version of the manuscript.

**Funding:** This research received no external funding.

**Institutional Review Board Statement:** Not applicable.

**Informed Consent Statement:** Not applicable.

**Acknowledgments:** Special thanks are warranted to those who reviewed early drafts of this paper: Matthew Flummer, Jim Slagle, Thad Botham, Kelly Fitzsimmons Burton, Jacobus Erasmus, Tyson James, Scott Olson, Jason Combs, Braxton Hunter, Johnathan Pritchett, Peter Rasor, Eric Hernandez, Russ Stratton and Suzanne Stratton.

**Conflicts of Interest:** The authors declare no conflict of interest.

## Notes

1  As this project aims to demonstrate that the free-thinking position is superior to its determinist counterparts (atheistic and Calvinistic) regarding epistemic responsibility on both philosophical and biblical grounds, the reader should note that our paper is engaging in systematic philosophical theology.

2  Plantinga's argument in *Warrant and Proper Function* is not in the same immediate family as our argument since it does not discuss causal determinism or the importance of libertarian freedom. However, it does seem to be a near-cousin, and we would be remiss not to mention his important work.

3  The word *determinism* is often meant to imply that all things or events are universally determined. Thus, the concept of the word *exhaustive* is already implied when speaking of determinism (as opposed to a few things being determined). This can get confusing. Because of this confusion, we have found it helpful to be slightly redundant and describe this view as *exhaustive* divine determinism (EDD) because it is often the case that those affirming determinism do not apply determinism to their entire mental life; they typically only have physical actions in mind. We often find that those opposing libertarian freedom unwittingly steal from libertarian freedom in an attempt to make their case. Thus, we believe the redundancy of EDD is helpful when having these discussions. (William Lane Craig has referred to this same concept as *universal divine determinism* here: "Molinism vs. Calvinism", accessed 21 June 2022, https://www.reasonablefaith.org/writings/question-answer/molinism-vs.-calvinism).

4  A naturalist may or may not affirm that all things in the universe are causally determined given the possibility of genuine quantum indeterminacy. However, as we discuss below, if naturalism is true it is reasonable to infer that everything about humanity is either determined by prior causes which have also been causally determined themselves, or causally determined by prior causes that were random or a product of chance. Either way, if naturalism is true, it seems that antecedent conditions are sufficient to necessitate all things about humanity. Moreover, we take quantum indeterminacy to be epistemological, not ontological.

5  See Plantinga (2011, pp. 307–19). Plantinga writes: "Naturalism is the idea that there is no such person as God or anything like him; immaterial selves would be too much like God, who, after all, is himself an immaterial self" (319).

   One reviewer has suggested that Plantinga's definition of naturalism is limited in that it denies the supernatural, but fails to preclude a transcendent, even divine reality that is compatible with a naturalistic worldview. We confess that it is difficult to make sense of how a worldview can include the "transcendent" and "divine" while also being thoroughly naturalistic. However, we acknowledge the wide semantic range of the term "naturalism" among scholars and do not want to disparage idiosyncratic uses of the term. For this paper, we will simply restrict ourselves to discussing the type of naturalism recognized by philosophers such as Plantinga, Graham Oppy, and Paul Draper.

6  We recognize that within atheist and Calvinist communities, a panoply of positions on free will do exist. Among atheists, Sam Harris, for example, leaves no opening for libertarian freedom; Thomas Nagel, on the other hand, seems to leave the door ajar. Among Calvinists, Guillaume Bignon and Paul Helm (1993) proffer determinism; Oliver Crisp, Gregory Koukl, and Richard Muller leave libertarian freedom an open option. The overwhelming tendency for both atheist and Calvinist scholars, however, is to resist libertarian accounts of free will as we have defined the concept here. We will proceed with the understanding that determinism is a very common position within the atheist and Calvinist communities, but is not generalizable to the whole of those communities (Harris 2012).

7  We are not suggesting that one is free to choose to believe whatever one would like. For example, even if we were each offered a billion dollars to truly believe that we are actually octopi, we could not do it. Sure, we could lie in an attempt to get the money, but we would not really believe that we are actually octopi. However, if indirect doxastic voluntarism is true, a person is truly responsible for (at least some of) his beliefs or propositional attitudes in the sense that he can exercise libertarian freedom at various points in life. For instance, he can freely think and choose (1) what he will or will not consider, (2) how a particular

subject is to be viewed, (3) if he is open to a particular line of argumentation or not, and so forth. Moreover, one can freely choose to be open-minded, or to be closed-minded, to focus or not to focus, to be more careful, or not. For more about indirect doxastic voluntarism, see Moreland and Craig (2017, pp. 309–10), Logos.

One reviewer notes that our treatment here lacks discussion on losing free will due to mental sickness or so-called possession. We recognize the possibility of one's losing free will due to mental incapacitation of these sorts. Since the FTA focuses on the incompatibility of determinism and epistemic responsibility, these cases would indeed add inductive support for the intuition that the lack of libertarian freedom removes epistemic responsibility. However, since our primary goal consists of defending our argument on philosophical and theological grounds, we believe such discussion is best reserved for a future paper.

Similar to the above, another reviewer noted that our paper lacked discussion on biological determinism and the effects of one's environment on free will. While we admit that biological and environmental factors serve to delimit (sometimes quite significantly) one's capacity and options for epistemic responsibility, we nevertheless find that this discussion would fall under the *degree* of epistemic responsibility one has under various circumstances rather than the deeper philosophical question which is our primary focus here: the incompatibility of determinism and epistemic responsibility.

8    The phrase *worth wanting* was used by Daniel C. Dennett in his book, *Elbow Room: The Varieties of Free Will Worth Wanting*. (Dennet 1984).

9    We are concerned about a specific kind of deliberation worth wanting. We will call this: *the power to deliberate and infer better or true beliefs over false ones*. This power is not compatible with determinism. After all, if an omnipotent deity of deception, for example, causally determines Carl to affirm a false theological belief in a specific circumstance, then it is impossible for Carl to deliberate in order to infer a better or true theological belief in that same circumstance. Pereboom (2008, pp. 287–306, https://doi.org/10.1007/s10892-008-9036-9) has argued that to deliberate rationally among distinct and alternative actions to perform, among other conditions, one cannot be certain of what action they will perform. Pereboom believes this kind of deliberation is compatible with a *belief* in determinism. While this is interesting, it is a completely different topic that should not be conflated with what we are arguing here. We believe the interesting point is not regarding a deliberation about *what to do* (or what action to perform), but rather, a deliberation about what one *ought to believe*. Does one have the power to infer better and true beliefs over false ones, or not? If EDD is true, for example, and God determines a Calvinist to affirm a false theological belief, then there is nothing the Calvinist can do to infer a better or true belief. The Calvinist in this scenario does not have the power to actively rationally deliberate and infer the truth. Rather, the Calvinist is a passive recipient of false beliefs, beliefs which he is deterministically caused to possess by God. We contend that in order to rationally deliberate and infer better and true beliefs, it would need to be the case that one's thoughts etc. are not determined by factors outside the agent's control. Moreover, if an agent is to *take himself to be deliberating*, then that agent must believe that his or her thoughts, intuitions, judgments, evaluations, and ensuing beliefs are not causally determined by something non-rational or someone who is untrustworthy. If this second claim is correct, then if a person comes to believe in meticulous determinism, that person could no longer construe certain mental processes as sequences of genuine deliberation, though one could still hold such sequences to be like what happens inside smart bombs or in some other inadequate, watered-down and highly counterintuitive compatibilist way.

10   Loosely speaking, an agent is a libertarian *free thinker* if (i) the entirety of the agent's mental activity is not determined by something or someone else, and (ii) the agent makes some mental choices.

11   The view that libertarian freedom necessarily requires alternative possibilities is losing popularity among prominent philosophers today. Elenore Stump describes this idea in "Moral Responsibility without Alternative Possibilities", chapter 8 of the book edited by David Widerker and Michael McKenna, *Moral Responsibility and Alternative Possibilities: Essays on the Importance of Alternative Possibilities* (Stump 2006). William Lane Craig is clear that the libertarian need not affirm alternative possibilities: "I am explicitly a libertarian about freedom of the will, and so there should be no doubt about that. I just deny the so-called Principle of Alternative Possibilities, that the ability to do otherwise in a given situation is a necessary condition of libertarian freedom." (Craig 2018). Although we do argue that humans occasionally possess alternative possibilities, arguing for the weaker version of libertarian freedom is sufficient. As this essay demonstrates, the FTA can be defended with or without the assumption of alternative possibilities.

12   Although it must be true that if one possesses strong libertarian freedom, then they necessarily possess weak libertarian freedom, some believe that if one possesses weak libertarian freedom, then they must possess strong libertarian freedom. This debate is beyond the scope of this project. Our concern is simply to show that based upon the definition of libertarian freedom offered, strictly speaking, alternative possibilities are not required (even if they are required in a broad metaphysical sense).

13   See Timothy A. Stratton's argument demonstrating that 1 Cor 10:13 implies libertarian freedom in *Human Freedom, Divine Knowledge, and Mere Molinism: A Biblical, Historical, Theological, and Philosophical Analysis* (Stratton 2020, p. 181).

14   To be more precise, weak libertarian freedom is a necessary condition for libertarian freedom. That is, a person possesses libertarian freedom only if he has the ability to think such that antecedent conditions are insufficient to causally determine or necessitate his thoughts and beliefs. We can state the libertarian necessary condition (LNC) in the following manner: LNC: *An agent, S, performs action, A, freely only if S is not causally determined to A.* Strong libertarian freedom, on the other hand, is a libertarian sufficient condition (LSC). That is, if a person has the opportunity to exercise an ability to choose between at least two options (including the options to refrain from action), each of which is compatible with his nature in a specific circumstance where the antecedent conditions are insufficient to causally determine or necessitate the agent's choice (thank you, Matthew

Flummer). Robert Kane (see *Four Views on Free Will*, Fischer et al. 2007, p. 15) makes a distinction between direct and indirect freedom pertaining to "self-forming actions" (SFAs). He thinks that we can be causally determined to act in some cases and still be free. For instance, suppose Bo knows that he is weak-willed and Bo does not want to succumb to temptation. Bo also knows that tonight he will be faced with temptation to sin. So, earlier in the day, Bo sets up a deterministic process in motion that ensures that he will withstand the temptation. If Bo had libertarian freedom when he set the process in motion, then it seems that Bo is free (or at least morally responsible in a desert sense) later even though Bo was caused to act by a deterministic causal process. Thus, a distinction can be made between direct and indirect freedom. Being 'directly free' at some time t is a necessary condition for libertarian freedom and moral responsibility. We are discussing something quite similar regarding rational oughts (as opposed to the moral oughts of SFAs) and discuss the difference between direct and indirect doxastic voluntarism. We do not think that one is typically directly free to choose any old belief at any moment. However, we do think that one must possess free thinking in a libertarian sense at different points along the way in order to rightly be held epistemically responsible.

[15] Being responsible in a desert sense is referring to the idea that one *deserves* to be praised or blamed for one's epistemic or intellectual behavior. For example, should the Calvinist who writes a book concluding that none of his ideas, thoughts, and beliefs are up to him (including those found in his book) be praised? If not, why should anyone agree with the Calvinist? Should the philosopher who disagrees be blamed? If not, there seems to be nothing wrong with joining the philosopher in disagreeing with the Calvinist.

[16] One might object that "this is a compulsive view of determinism that most determinists (compatibilists) would reject today. It's not that something is preventing you, it's that the determinants and your belief-forming processes align with each other". We believe this does not get the determinist/compatibilist off of any hooks. If determinism is true, then antecedent conditions are sufficient to necessitate the belief-forming processes, their alignment/non-alignment, along with one's ensuing beliefs. If antecedent conditions necessitate a belief-forming process to necessitate belief X, then one is deterministically prevented from believing anything other than X.

[17] One might appeal to arguments like those presented by Alvin Plantinga on page 194 of his *Warrant and Proper Function* (Plantinga 1993), in an attempt to show that rationality is consistent with determinism. That is, if one's beliefs are produced by properly functioning cognitive faculties in an environment suitable for those cognitive faculties according to a design plan successfully aimed at truth, then one can trust their beliefs. This approach fails if married to EDD. If EDD is true (emphasis on the *E*), then God has determined one's faculties to reach false metaphysical and theological beliefs. After all, no one is infallible. Below, we argue that proper function entails libertarian freedom when it comes to these issues. Moreover, if God determines exactly *how* one reasons, whenever they reason incorrectly, they reason exactly the way God determined them to reason and reach false conclusions about reality.

[18] (Real Atheology 2017), "Interview: Evan Fales", 42:28 to 42:37.

[19] Searle (2001, p. 202); emphasis in the original. Except for some interpolations added in brackets and a few ellipses—unless otherwise stated—no quotation in this essay has been edited or altered in any way.

[20] Searle, *Rationality in Action*, 66–67. It is vital to note that God is the perfect standard of knowledge. He does not approximate to the perfect standard, nor does he reason to better and true beliefs like humans do. On our model, the manner in which God knows things, unlike humans, is not through discursive reasoning. Humans, however, possess the power to approximate to God's perfect nature. With this in mind, we can say that to a degree that a human approximates to God's perfect standard of knowledge, to that same degree one is rational. To a degree that one fails to approximate to God's perfect knowledge, to that same degree a human is irrational. God does not infer best explanations over time. God is simply omniscient. So, the word *rational* does not apply to God in a deliberative sense. In another sense, God is rational in that he holds no false or contradictory beliefs and for all truths T, God believes T. It depends upon the sense and meaning of the word *rational*.

[21] The following definitions—taken from Van Inwagen (2017), p. 152—are especially important for this discussion:

- Hard determinism is the conjunction of determinism and incompatibilism. (Hard determinism thus entails the denial of the free-will thesis.)
- Soft-determinism is the conjunction of determinism and the free-will thesis [though the compatibilist and the libertarian understand the free-will thesis quite differently]. (Soft-determinism thus entails compatibilism.) It's vital to note that the FTA attacks determinism in general (both hard and soft).

[22] In naturalist William Rowe's authoritative assessment of Thomas Reid's arguments for libertarian agency, Rowe notes that according to Reid, the sort of consideration we have just presented entails that sooner or later, every event has in its ancestry a libertarian agent-cause. Setting aside the issue of uncaused events, if this is true, then it has clear theistic implications. Curiously, Rowe's only rebuttal to Reid (aside from claiming that some indeterministic events may occur with no causes), is the bald-faced assertion that it is coherent to think that atheism can be preserved if we hold that behind every event there must be either an agent cause or a physical cause. But this response completely overlooks the problem if transitivity of chains of physical causes. It won't do to respond by claiming that matter in Reid's day was conceived ultimately as passive corpuscles, but now, we depict matter as having causal forces/powers. This solves nothing because all such forces/powers are passive caused causes whose causal power must be triggered by a cause before it is actualized. Such *relata* still form transitive chains, so nothing is gained by this move. See Rowe (1991), pp. 54–57; see also pp. 94–121.

23    Moreland and Craig, *Philosophical Foundations*, 66.

24    "Justified true belief" is the traditional definition of *knowledge*, but recently it has been recognized by philosophers to only approximate it at best. However, most would say true belief by itself is insufficient for knowledge, so we need to add other elements, and these additions often involve some species of epistemic responsibility.

25    To clarify, the person in this deterministic circumstance does not perform an action. Rather, something else (i.e., a causal chain of events) performs an act upon the person. That is to say, a *subjective sensation of performing upon* passes through the person as opposed to the person performing upon something else.

26    Searle, *Rationality in Action*, 202. As an aside: if nothing we do *makes a difference*, then life seems meaningless. A meaningless life is definitely not worth wanting or having.

27    The fact that there seems to be a "Four Views" book on virtually every theological issue is evidence that Christians are not infallible.

28    For more regarding guidance control, see (Fischer and Ravizza [1998] 2008).

29    Matthew Flummer noted (via personal email, 7-31-22) that, strictly speaking, there might be indeterministic universes where no one has libertarian freedom but everything just randomly occurs. In such universes, although antecedent conditions are not sufficient to necessitate one's thoughts and actions, no one is in control of, or responsible for, anything. Something more is required and offered in the conclusion of this paper.

30    Jesus was quite concerned with not only our physical actions, but even more so regarding our mental thought life. For example, Jesus was clear that thinking lustfully about a woman was just as bad as physically committing adultery. Moreover, Jesus taught us to love God with all of our minds. If all of our thoughts are passive thoughts, we are not responsible (in a desert sense) for what passes through our minds. The apostle Paul strongly suggests that Christians (at least on occasion) have active control over what we think of and about. He says that we can actually *take thoughts captive* in 2 Cor 10:5 and warns that bad thinking can *take us captive* in Col 2:8.

31    A passive sensation is one in which a person has no control. If one finds themselves in an airplane and it suddenly drops in altitude because of turbulence, one experiences the sensation of butterflies in his stomach. This sensation was caused and determined by things other than the person and passed through the person (as it were)—causally determined via antecedent conditions. One has no choice but to experience what passes through him.

32    Slagle, *Epistemological Skyhook*, 25. (Slagle 2016).

33    Burton (2019), p. 23, Kindle. The word *deterministic* has been added in brackets with permission from Burton. This is because, although Burton's original examples all include things that imply malfunction in or manipulation of one's cognitive system, the point remains that these things are all non-rational or untrustworthy.

34    A passive cog is not responsible (in a desert sense) for what a deity of deception causes and determines it to think, how to judge and evaluate, and ultimately what to believe.

35    Slagle, *Epistemological Skyhook*, 206. (Slagle 2016).

36    Appealing to epistemic externalism does not escape the problem of exhaustive determinism. For example, if the mad scientist or a god of mischief determined our belief-forming faculties to get many things right, but also determined our belief-forming faculties to get many important things wrong (i.e., metaphysical and theological beliefs) and determines us to affirm said false beliefs, then we have a defeater to many more of our beliefs (even those that happen to be true). After all, we are in no epistemic position to know what beliefs we have been determined to affirm as true are actually false. Assuming that our faculties are typically reliable does not escape this problem. According to the externalist, beliefs are innocent until proven guilty. But why should a belief be *presupposed* as innocent if said belief is causally determined by something non-rational or someone who is untrustworthy? If one assumes his cognitive faculties are reliable because they were intelligently designed by a deity, the question is raised: What is this deity like? One must presuppose that God is anything but a "deity of deception" or a "god of mischief" who causally determines all of the deity's committed followers to affirm and advance false beliefs about reality. If one affirms EDD, then the determinist ultimately destroys the exact thing he is trying to explain—human rationality. If one presupposes that the Bible is reliable but then interprets Scripture in such a way that leads to the conclusion that the Author of Scripture is a deity of deception, then something has gone terribly wrong. On this view, there is reason to doubt our beliefs (regardless of whether they are true or false). This will be discussed further below.

37    This point is attributed to Jim Slagle, personal correspondence, 27 June.

38    One might counter and say this point has nothing to do with determinism, but rather, these examples imply malfunction in the cognitive system. This objection misses the point. We are pointing out the problems when thoughts and beliefs are determined by a non-rational process or an untrustworthy agent.

39    For the best defense of the Knowledge Argument, see Howard Robinson, *From the Knowledge Argument to Mental Substance* (Cambridge: Cambridge University Press, 2016). Cf. J. P. Moreland, "The Knowledge Argument Revisited", *International Philosophical Quarterly* 43 (June 2003): 219–28.

40    A special thanks to Thad Botham for suggesting this wording.

41    If one is not epistemically responsible for rejecting premise B2, then one has a defeater against his rejection of premise B2.

42    By epistemic responsibility, we are referring to the idea of basic desert (the biblical notion that one deserves to be praised or blamed for *how* they think and *what* they ultimately believe). We believe that humanity does possess the strong (LSC) sense of libertarian freedom. We will explain why this is the case. However, as will be demonstrated, one can defend this argument solely by addressing the weak (LNC) version of libertarian freedom. Ultimately, this paper will show that the Free-Thinking Argument can be defended with either the strong or weak sense and without equivocating between the two ideas from one premise to another.

43    A divine determinist might counter that perhaps God has good or morally sufficient reasons to determine Pastor Jones to believe the falsehood. This is ultimately irrelevant as God is still untrustworthy in the sense that he causally determines his own followers to affirm falsehoods about God and reality (even if it is for morally sufficient reasons).

44    Thad Botham (personal correspondence, 3 May 2022) shared that God's guaranteeing (by deterministically causing, for example) that we believe lots of falsehoods, *ceteris paribus*, seems wrong (a sort of problem of evil special case). Consider three situations:

1a    If Tina were placed in complete causal circumstances C, then C deterministically causes Tina to have false belief p.

2a    If Tina were placed in complete causal circumstances C*, non-determining with respect to whether or not Tina has false belief p, then it would be the case that Tina believes falsehood p. (Counterfactual of creaturely free epistemic situation.)

3a    If Tina were placed in complete causal circumstances C*, non-determining with respect to whether or not Tina has false belief p, then it would be the case that Tina does not believe falsehood p. (Counterfactual of creaturely free epistemic situation.)

Claim 3a is a necessary truth, and so God must believe 3a via God's natural knowledge. we think there's a compelling case for thinking that, *ceteris paribus*, if given the option of placing Tina in C vs. placing Tina in C*, it would be divinely preferable (again, *ceteris paribus*) to place Tina in C* whether 2a or rather 3a is true. If God believes 2a via middle knowledge (MK) and places Tina in C*, then Tina still had the opportunity in the same circumstances C* not to believe falsehood p. And if God believes 3a via MK and places Tina in C*, then Tina gets to avoid believing falsehood p. In short, God's placing Tina in C, where C necessarily guarantees that Tina believes a falsehood is, *ceteris paribus*, worse than Tina's believing falsehood p in non-determining circumstances C*, because these circumstances still left an opportunity for Tina not to believe falsehood p. Believing a falsehood with opportunity not to believe that falsehood is, *ceteris paribus*, better than being forced to believe a falsehood with no opportunity not to believe this falsehood. And, if EDD holds in our world, clearly there are many cases similar to 1a—none of us is infallible.

45    This seemingly growing number of Calvinists rejecting EDD is anecdotal and simply based upon people we have personally talked to. We must admit that although it seems that Calvinists rejecting EDD is a "growing number" it is still a small percentage.

46    Robust naturalism is the idea that only physical things exist. A fainthearted naturalist, by contrast, is one who helps herself to non-physical or immaterial emergent properties.

47    Moreland, "Theism", pp. 230–31. (Moreland 2018).

48    Van Inwagen (2013), p. 115; unnecessary capitalization has been removed from quotations of this source for stylistic reasons.

49    Inwagen, "Lewis's Argument Against Naturalism", p. 115. (Lewis 2011).

50    Plantinga, *Where the Conflict Really Lies*, p. 319.

51    Obviously, God is more than just a "thing"; he is the maximally greatest *Being*. However, we are using this admittedly unorthodox description of God for practical purposes.

52    For more regarding the Kalam cosmological argument and the fine-tuning argument, see Craig (2008).

53    "God made human beings in his own image" (Gen 9:6 NLT; see also Gen 1:27 and Jas 3:9). Hereafter, then, the phrase *things like God* must be understood in the light of these verses; that is, things made in God's own image.

54    Other supernatural kinds of things could exist like immaterial abstract objects. By definition, abstract objects—if they exist—do not have causal power. Although one might be inclined to agree with William Lane Craig and take a non-realist approach to *abstracta*, if one is wrong—and abstract objects do exist—unlike God and things like God, abstract objects would not have causal powers and would not be thinking kinds of things. On the other hand, there are concrete, unthinking things, like rocks and water molecules. These are like God in the sense that they are concrete objects, but unlike God in the sense that they are unthinking and material things. Ultimately, the term "like God" in our argument is referring to the fact that God is an immaterial, thinking thing, who possesses active causal power as a first mover. While we believe the human soul is an immaterial, thinking thing, who possesses active causal power as a first mover, we do show how Christians physicalists, who reject the idea of a soul, can still affirm this argument.

55    This new conversation is intended to get people into Scripture (we are pastors at heart).

56    Moreland, "Theism", p. 221. (Moreland 2018).

57    See notes 56 above.

58    Not only do God or things like God not exist in robust naturalism, if the physical universe is all that exists, then non-physical things like thoughts, aboutness, reasons, and thus, rationality—not to mention rationally inferred knowledge—do not exist either.

59    The view that human nature is a soul created in the likeness of God and possesses a physical body.

60　We believe libertarian freedom is a supernatural power since it is the ability to think and act such that the laws and events of nature do not always necessitate one's thoughts and actions. Thus, if one is 'free' from the laws and events of nature, it seems reasonable to infer that one is 'other than nature' and thus, supernatural (in some sense).

61　Moreland writes: "In my view, 'emergence' is just a name for the problem to be solved (how could simple emergent properties and substances emerge if you start with particles as depicted by physics and just rearrange them over time?). Among other things, this means that without some pretty serious, wildly ad hoc adjustments, the sort of unity possessed by consciousness (and, perhaps, its ground) cannot be located or otherwise explained, given robust or strong naturalism" ("Theism", 231).

62　The externalist might counter that this a valid assumption. The problem of self-defeat explained in this paper, however, provides a reason to reject this assumption.

63　William Lane Craig describes this recognition as a sense of vertigo for the determinist (see "Molinism vs. Calvinism"). Vertigo is a sudden internal sense of spinning. It seems that Craig is referring to an internal realization of reasoning in a circle.

64　As an analogy, consider the marksman and his rifle at a competition. His rifle was created—by an intelligent designer—to hit the bullseye if handled properly. Thus, the one wielding the rifle must make sure to follow many rules. Not only did he exercise freedom along the way while training for the competition, but while aiming at his target he must continue to carefully handle his weapon (lest he miss the target, be disqualified, or something far worse). Similarly, human cognitive faculties were created and intelligently designed by God to be able to hit the target of many metaphysical and theological truths if one is careful to take bad thinking captive before bad thinking takes them. Humans possess the power to aim at truth, but we must be careful, take thoughts captive, and judge with precision.

65　Moreland, "Theism", p. 220.

66　Moreland, "Theism", p. 238.

67　Hart (2016), p. 248. Hart notes that Paul Helm—the leading Calvinistic philosopher today—is a theological determinist (Hart, "Calvinism", 248n1). We must add, however, that there are some Calvinists who affirm all five points of TULIP and still affirm the libertarian freedom to choose between at least two options each compatible with one's nature in a certain circumstance. Gregory Koukl endorses this position (*Tactics*, pp. 175–76) and Oliver D. Crisp seems to affirm this possibility (Crisp 2014, pp. 71–96). Crisp notes that the Reformed theologian is "not necessarily committed to hard determinism" (*Deviant Calvinism*, p. 76), since it allows for "free will in some sense" (77). Indeed, although he says that "libertarian Calvinism [affirms that God] ordains whatever comes to pass" (87), Richard A. Muller adds that God "does not either determine or cause all things: some human acts are merely foreseen and permitted" (Muller 2017, p. 30, Kindle). This Calvinist/Reformed view is quite compatible with our view of Molinism.

68　This would be the case even if one thinks that epistemic externalism is true.

69　All theists surely agree that this description sounds more like—and most accurately applies to—Satan, "the father of lies" (John 8:44).

70　Thanks to Jacobus Erasmus for helping to craft this syllogism.

71　"Any Christian C" refers to a mature believer. This would exclude a toddler who, for example, might only affirm two theological beliefs that both happen to be true. This would include (but not be limited to) those who hold doctorates in philosophy or theology.

72　Moreland and Craig, *Philosophical Foundations*, p. 66.

73　See note 35 above.

74　John M. DePoe, online conversation, 23 June 2022.

75　Again, one might claim that this premise is false because a deity of deception requires the deity to be morally deficient; perhaps this deity has good reasons to deceive all Christians. As noted above, although we think this is unlikely, in this odd and ad hoc view, God could have morally sufficient reasons to be a deity of deception, but God would still be a deity of deception. No epistemic progress has been made.

76　Not only does this argument refrain from referencing alternative possibilities, but it provides an example of how one can defend the free-thinking argument by simply noting the epistemic problem of determinism. That is, if a human is not epistemically responsible for his or her thoughts and beliefs, then non-rational or untrustworthy deterministic causes are ultimately responsible for a human person's thoughts and beliefs.

77　Regarding randomness or luck, we believe that a Christian view of God and man—especially with Molinism in mind—does the best job of diminishing any "luck problems". If all things about an agent are causally determined by something or someone else (which seems to follow in both naturalism and EDD), then one is "lucky" if the external deterministic force beyond one's control causally determines one to affirm a true belief. If one's belief truly is random—and the agent happens to affirm a true belief—he is also lucky. However, if one is a rational agent created in the image/likeness of God, everything changes. If a human possesses the God-given powers of reflective self-control (see Franklin 2018), then—if one is careful to take thoughts captive and does his due diligence (as opposed to being lucky), then he *can* rationally infer and rationally affirm better and true beliefs in the actual world. This is perfectly compatible with Molinism. According to EDD, however, God does not choose the elect based upon anything about the individual—"lest any man should boast" (Eph 2:9 RSV). So, those who are "passed by" (the damned) and

those who are "elect" seem to be based upon luck or chance. Indeed, the elect seem to have won a cosmic and infinite lottery of sorts. Moreover, those who have been created for the sole purpose of eternal damnation are literally the unluckiest folks in all of creation. Ultimately, it seems we can find luck problems for any view. Some luck problems are worse than others.

78　　Some readers may be led to think that this reference to our being intelligently designed implies a conflict with the evolutionary development of humanity. However, this would be to conflate the *scientific* theses of common ancestry/random mutation/natural selection with the *metaphysical* claim that the mechanisms of evolution are unguided. However, as a scientific thesis, evolution cannot show that its requisite mechanisms and processes are not guided by a transcendent intelligence.

79　　Option (iv) is heavily inspired by Alvin Plantinga's description of when a belief has warrant (Plantinga 2015, p. 28, Logos). We like Plantinga's description, but believe it is incomplete. Indeed, it cannot be married to EDD. After all, if God determines all things, there is never truly any "dysfunction" in the mind of humans. Every incoherent thought and false belief would occur exactly as a deity of deception designed and determined. According to EDD, when one thinks poorly and believes incorrectly, one is functioning properly (from God's perspective). As an analogy, I (Stratton) often seek to learn how to handle my AR-15 responsibly, accurately, and tactically. My instructor, a former Green Beret, loads "dummy rounds" into our magazines ensuring that our rifles will malfunction during combat training (which requires the students to be able to fix the problem under stress and get back into the action as quickly as possible). But did my rifle really malfunction? Not from the Green Beret's perspective. This so-called "malfunction" occurred exactly as he intelligently designed, intended, and determined to take place. Malfunction or dysfunction did not really occur.

80　　To be exhaustive, we could include option (v): our theological beliefs are determined by a trustworthy God. This is a possibility, but it would not be one that we could know since the belief that our theological beliefs are caused by a trustworthy God could be determined by an untrustworthy God. Moreover, and more importantly, it flies in the face of our argument that a trustworthy God would not determine false theological beliefs. Combined with the fact that all humans with theological beliefs (including all Christians) possess at least one false theological belief, option (v) is ruled out.

81　　A special thanks is warranted to Tyler Dalton McNabb for recommending a conditional move (via personal correspondence, 1 August 2017).

82　　Moreland (2014), p. 53. Consider also the following deductive argument adapted from Moreland, *Soul*, pp. 125–26.

1b　　The law of identity is true: If x is identical to y, then whatever is true of x is true of y and vice versa.
2b　　I can strongly conceive of myself as existing disembodied.
3b　　If I can strongly conceive of some state of affairs S that S possibly obtains, then I have good grounds for believing that S is possible.
4b　　Therefore, I have good grounds for believing of myself that it is possible for me to exist and be disembodied.
5b　　If some entity x is such that it is possible for x to exist without y, then (i) x is not identical to y, and (ii) y is not essential to x.
6b　　My body (or brain) is not such that it is possible to exist disembodied, i.e., my body (or brain) is essentially physical.
7b　　Therefore, I have good grounds for believing of myself that I am not identical to my body (or brain) and that my physical body is not essential to me.

　　　　See also, Moreland (2009).

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
