# Peer review of "An Explanation and Defense of the Free-Thinking Argument"

_religions, doi:10.3390/rel13100988_

Round 1

Reviewer 1 Report

In my view, the author should up front that they are engaged in philosophical theology, because unapologetically takes a definite (and sensible) Christian theological position when both countering atheists and 'extreme' Calvinism.

Perhaps he should point out (perhaps in a footnote) that atheists and Calvinists can take different views on free will. Caveats of relativities should be better documented.

I missed discussion of losing free will in mental sickness or so-called possession. Scholars like John Wisdom (the greater the madness the less free the will) and Gregory Bateson (on the ecologies of mentally disturbed minds) get no look in.

Thanks

Garry WT

Author Response

Thank you for your feedback! Please see our responses below:

Point 1: In my view, the author should up front that they are engaged in philosophical theology, because unapologetically takes a definite (and sensible) Christian theological position when both countering atheists and 'extreme' Calvinism.

Response to Point 1: We have included the following as a footnote on page 1:

As this project aims to demonstrate that the free-thinking position is superior to its determinist counterparts (atheistic and Calvinistic) regarding epistemic responsibility on both philosophical and biblical grounds, the reader should note that our paper is engaging in systematic philosophical theology.

Point 2: Perhaps he should point out (perhaps in a footnote) that atheists and Calvinists can take different views on free will. Caveats of relativities should be better documented.

Response to Point 2: We have included the following as a footnote on page 2:

We recognize that within atheist and Calvinist communities, a panoply of positions on free will do exist. Among atheists, Sam Harris (“The Illusion of Free Will,” 2012), for example, leaves no opening for libertarian freedom; Thomas Nagel (“Free Will,” 1987), on the other hand, seems to leave the door ajar. Among Calvinists, Guillaume Bignon and Paul Helm proffer determinism; Oliver Crisp, Gregory Koukl, and Richard Muller leave libertarian freedom an open option. The overwhelming tendency for both atheist and Calvinist scholars, however, is to resist libertarian accounts of free will as we have defined the concept here. We will proceed with the understanding that determinism is a very common position within the atheist and Calvinist communities, but is not generalizable to the whole of those communities.

Point 3: I missed discussion of losing free will in mental sickness or so-called possession. Scholars like John Wisdom (the greater the madness the less free the will) and Gregory Bateson (on the ecologies of mentally disturbed minds) get no look in.

Response to Point 3: We have included the following as a footnote on page 3:

One reviewer notes that our treatment here lacks discussion on losing free will due to mental sickness or so-called possession. We recognize the possibility of one’s losing free will due to mental incapacitation of these sorts. Since the FTA focuses on the incompatibility of determinism and epistemic responsibility, these cases would indeed add inductive support for the intuition that the lack of libertarian freedom removes epistemic responsibility. However, since our primary goal consists of defending our argument on philosophical and theological grounds, we believe such discussion is best reserved for a future paper.

Reviewer 2 Report

A defense of the overarching assumptions that underpin the free-thinking argument may be found in this article. The purpose of this line of reasoning is to show that determinism and epistemic responsibility, in the desert meaning, are incompatible with one another (being praised or blamed for any thought, idea, judgment, or belief). The article concludes that if one believes that they are a rational free thinker who is not ultimately mind-controlled by something (or someone) else, then they should reject the determinism that seems to follow from both naturalism and EDD. Instead, one needs to assert that there is a God who operates outside of natural laws. In addition, one should be conscious of the fact that he or she has a supernatural, immaterial, active, rational, and free-thinking creature known as a soul, which was made in the image and likeness of God and will continue to exist after the death of one's physical body.

I think the discussion is great. But there is also another question I’d like to hear the author discuss and address at the beginning, which is biological determinism and the faith of biology. For example, even if we are solely responsible for our thoughts, does that mean free will? Will it be that we are genetically this way? Why is A more diligent than B? Did A choose to be more diligent? Or was A “wired” this way? If A is more successful than B, is it pre-determined? A is genetically better and in a better nurturing environment. Even if A is in a bad environment, A wins the genetic lottery to have some good traits in him to be successful? Does real free will exist?

Please consult the following books and articles:

Pollack, Robert. 2000. The Faith of Biology & the Biology of Faith : Order, Meaning, and Free Will in Modern Medical Science. New York: Columbia University Press.

Summerfield, Scott G, and Phil Jeffrey. 2006. “In Vitro Prediction of Brain Penetration - a Case for Free Thinking?” Expert Opinion on Drug Discovery 1 (6): 595–607. https://doi.org/10.1517/17460441.1.6.595.

Tattersall, Ian, and Robert DeSalle. 2019. The Accidental Homo Sapiens. New York: Pegasus Books.

Willmott, Chris. 2016. Biological Determinism, Free Will and Moral Responsibility. Cham: Springer International Publishing AG.

Tendler, Moshe David. 2011. “Behavioral Genetics and Free Will.” B'or ha'Torah, no. 21: 35–43.

Alper, Joseph S. 1998. “Genes, Free Will, and Criminal Responsibility.” Social Science & Medicine (1982) 46 (12): 1599–1611. https://doi.org/10.1016/S0277-9536(97)10136-8.

Pollack, Robert. 2013. The Faith of Biology & the Biology of Faith : Order, Meaning, and Free Will in Modern Medical Science. New York: Columbia University Press.

Author Response

Thank you for your feedback! Please see our response below:

Point 1: I think the discussion is great. But there is also another question I’d like to hear the author discuss and address at the beginning, which is biological determinism and the faith of biology. For example, even if we are solely responsible for our thoughts, does that mean free will? Will it be that we are genetically this way? Why is A more diligent than B? Did A choose to be more diligent? Or was A “wired” this way? If A is more successful than B, is it pre-determined? A is genetically better and in a better nurturing environment. Even if A is in a bad environment, A wins the genetic lottery to have some good traits in him to be successful? Does real free will exist?

Please consult the following books and articles:

Response to Point 1: We have included the following in a footnote on page 3:

Similar to the above, another reviewer noted that our paper lacked discussion on biological determinism and the effects of one’s environment on free will. While we admit that biological and environmental factors serve to delimit (sometimes quite significantly) one’s capacity and options for epistemic responsibility, we nevertheless find that this discussion would fall under the degree of epistemic responsibility one has under various circumstances rather than the deeper philosophical question which is our primary focus here: the incompatibility of determinism and epistemic responsibility.

Reviewer 3 Report

The paper is well written and, for the most part, coherent. However, there are some serious disconnections/inconsistencies that I think make it unworthy for publication. The following are some examples:

-The author seems, in places, to conflate naturalism with determinism, which is certainly not necessary.

-The author cites Plantigua's definition of naturalism as denying God or anything like God. However, Plantigua's definition is rather limited. Naturalism simply denies the supernatural but, as some have argued, does not preclude positing some transcendent, even divine reality that is perfectly compatible with a naturalistic worldview. The author needs to at least recognize this but does not.

-The author cites a "Christian biblical perspective" as a justification for the Free Thinking Argument and cites I Cor. multiple times as support. I am not sure that a biblical perspective can be adequately used in an argument for or against philosophical determinism to provide any type of credible evidence. Furthermore, citing a biblical reference in support of FTA fails to recognize that the Bible can just as easily be used to support a determinist perspective and has many times in the past (e.g., forms of strict Calvinism). However, if the author wants to make a strong philosophical argument, picking a verse that happens to support the favored position weakens the philosophical strength of the argument. 

-The author mistakenly claims E. O. Wilson is a determinist. Wilson does not refute freedom of will and human choice. Instead he discounts philosophical justifications for the origins and function of human thought that are antiquated and have long since been disproven by modern science even though some modern philosophers continue to throw them out as if they are still relevant. 

-I find the use of the "biblical view of God" as a credible explanation for human libertarian freedom and the human soul to be dubious at best.

The inconsistent use and explanation of terminology hurt the overall argument. Furthermore, while the author strongly states that a person's metaphysical and theological beliefs are clearly the result of option 4 (the existence of a free-thinking, perfect, and supernatural God), the philosophical arguments are not strong in proving this point. Additionally, the reference to humans as "intelligently designed" free-thinking agents seems to imply a conflict with the evolutionary development of humanity, which severely weakens any argument on human thought since it either ignores or discounts the best available science on the evolutionary development of human thought.

While I agree in principle with the author's conclusion concerning the justification of the FTA argument over determinism, I do not think that the article in its current form is worthy of further consideration for publication. 

Author Response

Thank you for your feedback! Please see our responses below:

Point 1: The author seems, in places, to conflate naturalism with determinism, which is certainly not necessary.

Response to Point 1: We disagree that we have conflated naturalism with determinism. The following are specific examples from the paper where we distinguish between the two:

A naturalist may or may not affirm that all things in the universe are causally determined given the possibility of genuine quantum indeterminacy. (fn 7, p. 2)

The reason, however, why most naturalists reject libertarian freedom is because libertarian freedom does not seem to fit nicely with most accounts of naturalism… (p. 8)

Note that this argument says nothing directly about naturalism, atheism, materialism, physicalism, Calvinism, Molinism, or any other “-ism” that might be relevant to this discussion. However, speaking of “isms,” this argument does conclude that humanity possesses the libertarian freedom to think. If that is true, then we have indirectly demonstrated that incompatibilism is true. That is to say, epistemic responsibility is not compatible with determinism. (p. 18-19)

If one believes that he or she is a rational free-thinker who is not ultimately mind-controlled by something (or someone) else, then one should reject the determinism that seems to follow from both naturalism and EDD. (p. 29)

Point 2: The author cites Plantigua's definition of naturalism as denying God or anything like God. However, Plantigua's definition is rather limited. Naturalism simply denies the supernatural but, as some have argued, does not preclude positing some transcendent, even divine reality that is perfectly compatible with a naturalistic worldview. The author needs to at least recognize this but does not.

Response to Point 2: We have included the following in a footnote on page 2:

One reviewer has suggested that Plantinga’s definition of naturalism is limited in that it denies the supernatural, but fails to preclude a transcendent, even divine reality that is compatible with a naturalistic worldview. We confess that it is difficult to make sense of how a worldview can include the “transcendent” and “divine” while also being thoroughly naturalistic. However, we acknowledge the wide semantic range of the term “naturalism” among scholars and do not want to disparage idiosyncratic uses of the term. For this paper, we will simply restrict ourselves to discussing the type of naturalism recognized by philosophers such as Plantinga, William Lane Craig, Graham Oppy, and Paul Draper.

Point 3: The author cites a "Christian biblical perspective" as a justification for the Free Thinking Argument and cites I Cor. multiple times as support. I am not sure that a biblical perspective can be adequately used in an argument for or against philosophical determinism to provide any type of credible evidence. Furthermore, citing a biblical reference in support of FTA fails to recognize that the Bible can just as easily be used to support a determinist perspective and has many times in the past (e.g., forms of strict Calvinism). However, if the author wants to make a strong philosophical argument, picking a verse that happens to support the favored position weakens the philosophical strength of the argument.

Response to Point 3: Different types of justification will presumably appeal to different people. Many Calvinists, for example, are apprehensive about philosophical proofs, but are less apprehensive about biblical prooftexts and so will hopefully be challenged to view the cited support texts in a new light.

However, we are unclear how “picking a verse that happens to support a favored position weakens the philosophical strength of the argument.” Since none of the syllogisms invoke biblical prooftexts, either in the premises or in support of those premises, they stand independently of such considerations.

Point 4: The author mistakenly claims E. O. Wilson is a determinist. Wilson does not refute freedom of will and human choice. Instead he discounts philosophical justifications for the origins and function of human thought that are antiquated and have long since been disproven by modern science even though some modern philosophers continue to throw them out as if they are still relevant.

Response to Point 4: We do believe that E. O. Wilson was a determinist, though his determinism was couched in metaphor. However, since his view is not as clear as that of other determinists, we have removed references to him.

Point 5: I find the use of the "biblical view of God" as a credible explanation for human libertarian freedom and the human soul to be dubious at best.

Response to Point 5: We find the “biblical view of God” very credibly explains human libertarian freedom and the human soul. Unfortunately, proving the truth of Christianity and this explanatory power is outside the scope of this paper. We hope that non-Christians as well as Christians who reject the ability of “the biblical view of God” to explain libertarian freedom and the human soul will both be challenged by the FTA itself to reject determinism as incompatible with epistemic responsibility, which was the primary focus of the paper. Any further investigation it incites into the truth of Christianity and the resources it provides for libertarian freedom is a bonus.

Point 6: The inconsistent use and explanation of terminology hurt the overall argument. Furthermore, while the author strongly states that a person's metaphysical and theological beliefs are clearly the result of option 4 (the existence of a free-thinking, perfect, and supernatural God), the philosophical arguments are not strong in proving this point. Additionally, the reference to humans as "intelligently designed" free-thinking agents seems to imply a conflict with the evolutionary development of humanity, which severely weakens any argument on human thought since it either ignores or discounts the best available science on the evolutionary development of human thought.

Response to Point 6: Since no examples are given of “The inconsistent use and explanation of terminology,” we cannot make corrections on this point.

Regarding the claim that the philosophical arguments were not strong in proving option 4, we can only note that abductive inference is a known and viable form of reasoning. We listed the live options and, after giving reasons why the other options were not good explanations, concluded that option 4 was the best explanation.

Regarding the claim that reference to intelligent design implies a conflict with the evolutionary development of humanity, this would be to conflate the scientific theses of common ancestry/random mutation/natural selection with the metaphysical claim that the mechanisms of evolution are unguided. However, as a scientific thesis, evolution cannot show that its requisite processes are not guided by a transcendent intelligence. We have added the following footnote clarifying this potential misunderstanding:

Some readers may be led to think that this reference to our being intelligently designed implies a conflict with the evolutionary development of humanity. However, this would be to conflate the scientific theses of common ancestry/random mutation/natural selection with the metaphysical claim that the mechanisms of evolution are unguided. However, as a scientific thesis, evolution cannot show that its requisite mechanisms and processes are not guided by a transcendent intelligence.

Round 2

Reviewer 3 Report

The revised version of the paper does not change my initial assessment that this paper is not of sufficient quality for publication.